# Discordance to ASHP Therapeutic Guidelines Increases the Risk of Surgical Site Infection

**DOI:** 10.3390/ph14111088

**Published:** 2021-10-27

**Authors:** Fauna Herawati, Rika Yulia, Heru Wiyono, Firdaus Kabiru Massey, Nurlina Muliani, Kevin Kantono, Diantha Soemantri, Retnosari Andrajati

**Affiliations:** 1Department of Clinical and Community Pharmacy, Faculty of Pharmacy, Universitas Surabaya, Surabaya 60293, Indonesia; rika_y@staff.ubaya.ac.id (R.Y.); firdaus.kabiru@gmail.com (F.K.M.); nurlinamuliani95@gmail.com (N.M.); 2Department of Pharmacology and Clinical Pharmacy, Faculty of Pharmacy, Universitas Indonesia, Depok 16424, Indonesia; retnosaria@gmail.com; 3Faculty of Medicine, Universitas Surabaya, Surabaya 60293, Indonesia; herwyno@yahoo.com; 4Department of Food Science, Auckland University of Technology, Private Bag 92006, Auckland 1142, New Zealand; kevin.kantono@aut.ac.nz; 5Department of Medical Education, Faculty of Medicine, Universitas Indonesia, Depok 16424, Indonesia; diantha.soemantri@ui.ac.id

**Keywords:** defined daily dose, surgical site infection, antibiotic stewardship, hospital

## Abstract

Clean surgery without contamination does not require prophylactic antibiotics, but there are high-risk surgical procedures that may cause infection and/or involve vital organs such as the heart, brain, and lungs, and these indeed require the use of antibiotics. This study aimed to determine the quantity of antibiotic use based on the defined daily dose (DDD) per 100 bed days and the quality of antibiotic use based on the percentage of concordance with The American Society of Health-System Pharmacists (ASHP) therapeutic guidelines (i.e., route of administration, time of administration, selection, dose, and duration). This includes the profiles of surgical site infection (SSI) in surgical patients from January through June 2019. The study subjects were 487 surgical patients from two hospitals who met the inclusion criteria. There were 322 patients out of 487 patients who had a clean surgical procedure. Ceftriaxone (J01DD04) was the most used antibiotic, with a total DDD/100 bed days value in hospitals A and B, respectively: pre-surgery: 14.71, 77.65, during surgery: 22.57, 87.31, and postsurgery: 38.34, 93.65. In addition, 35% of antibiotics were given more than 120 min before incision. The lowest concordance to ASHP therapeutic guideline in hospital A (17.6%) and B (1.9%) was antibiotic selection. Two patients experienced SSI with bacterial growths of *Proteus* spp., *Staphylococcus aureus*, *Staphylococcus epidermidis*, and *Escherichia coli*. The usage of prophylactic antibiotics for surgical procedures was high and varied between hospitals. Hospital B had significantly lower concordance to antibiotic therapeutic guidelines, resulting to a higher infection rate, compared with hospital A. ASHP adherence components were then further investigated, after which antibiotic dosing interval and injection time was found to be a significant predictor for positive bacterial growth based on logit–logistic regression. Further interventions and strategies to implement antibiotic stewardship is needed to improve antibiotic prescriptions and their use.

## 1. Introduction

Antimicrobial resistance threatens global health; antibiotic use in surgical specialties is an infection control indicator and may reflect the incidence of surgical site infection. In most developing countries, particularly Indonesia, antibiotics are given before the incision but also during the time from admission to discharge from the hospital which may result in higher chances of antimicrobial resistance. A new strategy is therefore needed to overcome this problem.

Surgery is a medical procedure in the form of an incision made by a surgeon with the aim of preventing or treating an illness. In general, all surgical procedures can pose a risk of infection. Thus, antibiotic prophylaxis is needed to prevent postoperative infection. However, not all types of surgery require antibiotic prophylaxis. Clean surgery without contamination does not require prophylactic antibiotics unless there are high-risk procedures that may cause infection and/or involve vital organs such as the heart, brain, and lungs. Prophylactic antibiotics are generally recommended in some clean and all contaminated operations [1]. In general, the use of antibiotic prophylaxis must be in accordance with therapeutic standards. However, the percentage compliance with guidelines for antibiotic use (e.g., NICE Guidelines, Stanford Health Care Guidelines, Surgical Antimicrobial Prophylaxis Guidelines) is relatively low (under 50%) in several countries such as Qatar (46.5%), Pakistan (<50%), Italy (40%), and the Philippines (13%) [2].

The mismatch of antibiotic use with therapeutic guidelines is reported to be high in surgical ward [3], ICU [3,4], and medical ward [3] with mismatches to the indication, antibiotic selection [3,4], and time/dose/route/duration of antibiotic administration. Often, differences existed from the internal guidelines [3] especially in prescribing prophylactic antibiotics pre- and postsurgery. A study showed that in pre-surgery, 46% incorrect timing, 32% incorrect dosages, and 20% overly broad antibiotic spectrum were reported; while in postsurgery, they found 74% incorrect duration of antibiotic administration, 29% incorrect dosage or frequency, and 6% overly broad spectrum of antibiotics [5].

Prescribing antibiotics preoperatively can reduce the occurrence of wound infection. Whereas not following the guidelines, for example, by using different spectrum antibiotics or different tissue antibiotic concentrations at the start and during surgery, can reduce antibiotics’ effectiveness. Moreover, antibiotic selection pressure leads to antibiotic resistance. The incidence of SSI is significantly higher when prophylactic antibiotics are given postsurgery than when given presurgery with an odds ratio (OR) of 1.89 (95% CI 1.05–3.4). Prophylactic antibiotics more than 120 min preincision showed a higher prevalence of SSI than those within 120 min (OR 5.26; 95% CI (3.29–8.39)) [6,7,8]. The use of prophylactic antibiotics that are irrational or not in accordance with therapeutic guidelines requires preventive efforts so as not to cause cases of resistance to prophylactic antibiotics. According to the Indonesian National Action Plan for Antimicrobial Resistance 2017–2019, one of the preventive efforts that can be conducted is to evaluate the use of prophylactic antibiotics in its quantity and quality.

World Health Organization (2019) states that implementing antimicrobial stewardship by evaluating the use of antibiotics can optimize the use of antibiotics, encourage behavior change in prescribing antibiotics, improve quality of care and patient outcomes, save unnecessary health care costs, and reduce the emergence, selection, and further spread of antimicrobial resistance (AMR) [9,10,11,12,13,14]. A systematic review by Nathwani et al. (2019) of 146 studies in North America (49%), Europe (25%), Asia (14%), Africa (3%), South America (3%), and Australia (3%) reported a decrease in length of stay (LOS) and antibiotic expenditure, respectively, by 85 and 92%; mean cost savings in the US study were USD 732 per patient (range: USD 2.50 to 2640), with a similar trend shown in European studies [14]. Hence, taking into account the importance and positive outcome of antibiotics stewardship, this study aimed to evaluate the quantity (DDD/100 bed-days) and quality of prophylactic antibiotic use in surgical patients in hospitals while assessing the impact of concordance towards ASHP therapeutic guideline towards the risk of SSI.

## 2. Results

A total of 487 patients in both hospitals met the inclusion and exclusion criteria. Table 1 describes the demographic information of the patients in the study. Table 1 shows that 268 patients (55%) were female, 298 patients (61%) were aged between 31 and 60 years old, 393 patients (81%) of patients with out-of-pocket payment, and 57 patients (12%) had comorbidities. Overall, 322 patients underwent surgery with a clean operation category (94 patients in hospital A, and 228 patients in hospital B) (Table 1); in 164 patients (34%), the injection time given was greater than 120 min (Table 2).

### 2.1. Quantity Profile of Prophylactic Antibiotics

Pre-, During, and Postsurgeries

Based on the recapitulation of antibiotic prophylaxis (DDD/100 bed-days) in hospitals A and B, respectively, between January and June 2019, the antibiotic use prophylaxis presurgery was 44.26 and 133.58; during surgery was 33.32 and 132.4; postsurgery was 66.65 and 148.68. Ceftriaxone had the highest DDD/100 patient days in the pre-, during, and postsurgery, with sequences of 14.71 and 77.65, 22.57 and 87.31, 38.34, and 93.65 (Table 3). Overall, hospital B had significantly higher usage then hospital A (F_(1,5)_ = 66.124, *p* < 0.001).

### 2.2. Quality Profiles of Prophylactic Antibiotics

Other than the route of administration, compatibility with ASHP guidelines was less than 50%. Order from highest to lowest conformity, the concordance to the ASHP guidelines was route of administration, injection given time, dosage, dosing interval, and selection (Table 4). In general, hospital A was more compliant than hospital B. Logistic regression revealed that injection given time (X^2^_(1,162)_ = 283.62; *p* < 0.001) and dosing interval (X^2^_(1,162)_ = 4.165; *p* < 0.041) were significant predictors for infection cases in hospital B.

### 2.3. Profile of Surgical Site Infection (SSI)

Surgical site infection occurs related to surgical incision procedures within 90 days of follow-up. Only two patients experienced SSI in this study. Both patients experiencing postoperative SSI were identified after undergoing a third review in the first patient, and after the first review in the second patient. The results of the first patient’s pus culture showed the presence of *Proteus* spp. and *S. aureus*, while the second patient had *E. coli* and *S. epidermidis*. Profiles of the types of bacteria that cause SSI are shown in Table 5. Both SSI patients were known to have had a history of diabetes mellitus, with a random blood sugar level of <200 mg/dL while in the hospital, LOS preoperatively for >1 day, undergoing surgery with a duration of operation >60 min. The first SSI patient did not receive presurgical antibiotics, while the second patient received 2 × 1 g of ceftriaxone preoperative antibiotics for 1 day. Both SSI patients received the same prophylactic antibiotic, i.e., ceftriaxone 2 × 1 g with the administration time > 120 min, and both dosages were continued until 4–5 days after incision. Neither received antibiotics when they were discharged.

### 2.4. Bacteria Sensitivity Profile Causes SSI

The results of antibiotic sensitivity tests showed that *Escherichia coli* bacteria are still sensitive to several types of antibiotics, such as cefoxitin, chloramphenicol, fosfomycin, oxacillin, tetracycline, and vancomycin. Bacteria *Proteus* spp. were sensitive to antibiotics such as amoxicillin, amoxicillin–clavulanate, ampicillin, ciprofloxacin, cotrimoxazole, fosfomycin, norfloxacin, ofloxacin. *Proteus* spp. and had intermediate sensitivity to cefoxitin and chloramphenicol. *Staphylococcus aureus* bacteria were sensitive to chloramphenicol, oxacillin, and vancomycin and had intermediate sensitivity to amoxicillin, amoxicillin–clavulanate, and cefoxitin.

## 3. Discussion

### 3.1. Quantity Profiles of Prophylactic Antibiotics

The assessment of the quantity of antibiotic use was conducted by calculating DDD/100 bed-days based on the classification (Anatomical Therapeutic Chemical Classification) that has been approved worldwide by the World Health Organization (WHO). DDD is defined as the use of antibiotics based on the average daily dose given to 70 kgs adult patients according to main therapeutic indications. However, DDD cannot provide information pertaining to the suitability of the actual daily dose prescription. DDD calculation is influenced by the amount of antibiotic use (grams) and the total length of stay (bed-days). The higher percentage of antibiotics use leads to the greater the DDD value [15].

The calculation of DDD/100 bed-days between January and June 2019 was divided into pre-, during, and postsurgeries. Overall, the average of prophylactic antibiotic use in pre, during, and postsurgeries in hospital B was higher than in hospital A. Ceftriaxone is the most widely used antibiotic including surgery. Tolbert’s 2019 evaluative study on the use of ceftriaxone in tertiary hospitals in Tanzania showed 322 (51.1%) of 630 research samples used ceftriaxone antibiotics. A total of 44 samples (40.7%) received ceftriaxone for surgical prophylaxis purposes that were divided into 32 (72.7%) samples receiving antibiotics before surgery, 3 samples (6.8%) samples receiving ceftriaxone antibiotics during surgery, and 9 samples (20.5%) receiving antibiotics after surgery [16]. Another evaluative study on the use of prophylactic antibiotics in surgical patients at Nekemte Hospital Ethiopia showed that the 153 ceftriaxone samples were the most widely used antibiotics, with a total of 66 samples (43.1%) [17].

Prophylactic antibiotics can significantly reduce the risk of postoperative infections. A meta-analysis of 40 studies testing the effectiveness of ceftriaxone compared with the comparators as the surgical prophylaxis (ampicillin–sulbactam, benzylpenicillin, cefamandole, cefazolin, cefotaxime, cefuroxime) showed that in clean operations, the incidence of SSI in ceftriaxone was 5.14%, and comparator 6.2% (OR −0.22, 95% CI −0.51 to 0.01; *p* = 0.047). In clean-contaminated operations, the incidence of SSI in ceftriaxone was 4.6% and comparator 6.4% (OR −0.36, 95% CI −0.67 to −0.13; *p* = 0.001) [18].

The average value of DDD/100 prophylactic antibiotic bed-days in this study is higher than other studies in one hospital in Indonesia but is still lower than that in the study of the use of prophylactic antibiotics in one hospital in Turkey. The ceftriaxone antibiotics dosage used in hospital B was 87 DDD/100 bed-days, which means there were 87 out of 100 patients who received ceftriaxone 1 DDD antibiotics per day. A primary study of surgical patients in the hospitals in Surabaya assessed the total use of ceftriaxone prophylactic antibiotics using the DDD/100 bed-days unit, and the total use of these antibiotics was 30–54 DDD/100 bed-days [19,20]. These results are still lower when compared with DDD/100 bed-days of this study. However, the total use of prophylactic antibiotics in this study was still lower than that in Turkey, which was 132.4 DDD/100 bed-days, compared with 289.32 DDD/100 bed-days [21].

### 3.2. Quality Profiles of Prophylactic Antibiotics

According to Permenkes RI RI No. 2406/MENKES/PER/XII/2011 on General Guidelines for the Use of Antibiotics in 2011, the basis for selecting antibiotics should be drawn from the patterns of bacterial sensitivity to antibiotics, the clinical condition of patients, the selection priorities of first-line antibiotics or antibiotics with a narrow spectrum, cost, and adjusted according to the hospital diagnosis and therapy (PDT) Guidelines [11]. The results in this study indicated that of the 487 samples analyzed between January and June 2019, the percentage of compliance with ASHP therapeutic guidelines for the selection of antibiotic types was 18% in hospital A and 2% in hospital B. We postulate that the low accordance in hospital B may be due to the fact that in Indonesia, antibiotics selection depends on the availability of the drugs rather than guideline recommendation.

Based on ASHP therapeutic guidelines, to ensure adequate plasma and tissue concentration during surgery, it is recommended that the drug to be given intravenously within one hour of an incision. In this study, more than 95% of patients received prophylactic antibiotics intravenously, but only around 50% in concordance with the preoperative dosage timing recommendation.

Some antibiotics, such as vancomycin and fluoroquinolones, must be given 120 min before surgery due to the influence of the half-life and protein binding of each antibiotic [22,23]. Allegranzi [7] conducted a meta-analysis of 13 observational studies on 53,975 adult patients to assess the optimal time of administration of surgical prophylactic antibiotics and found that preoperative antibiotics are better than postsurgical drugs to prevent surgical site infection (SSI) (OR 1.89; 95% CI 1.05–3.4). Administration of antibiotics earlier than 120 min showed a higher prevalence of SSI than the administration of antibiotics within 120 min (OR 5.26; 95% CI 3.29–8.39).

The incidence of SSI in surgery patients in hospital B from January through June 2019 was 2 out of 323 patients (0.62%). The SSI patients in this study both had ceftriaxone at intervals and doses of 2 × 1 g. The administration time was more than 2 h preincision. Mismatched timing of prophylactic antibiotics can cause SSI and the growth of resistant bacteria. Research conducted by Billoro in 2019 to evaluate the use of prophylactic antibiotics in 255 surgical patients in southern Ethiopia, using a prospective cohort study, showed that administration of surgical prophylactic antibiotics earlier than 1 h before the incision was associated with 20% of SSI cases, whereas administration within 1 h had a lower incidence, at 11.4% (*p* = 0.012). In addition, antibiotic resistance was observed in most types of bacteria in the surgical ward. One of these bacteria, *Escherichia coli*, was resistant to the antibiotics such as ceftriaxone, chloramphenicol, ciprofloxacin, and gentamicin [24].

The selection of the right drug must be accompanied by an accurate dose determination. ASHP guidelines recommend ceftriaxone 2 g, or 50–75 mg/kg body weight, for adults [22,23]. The average body weight of the study sample was >40 kg; thus, the administration of <1 g of ceftriaxone injection was not in accordance with ASHP guidelines [25].

Aside from the timing of the first dose, the dosing interval and duration must be appropriate to achieve minimum effective plasma and tissue concentrations. ASHP recommends a single dose or that the duration of prophylactic antibiotics be <24 h postoperatively [23]. Using the case of more than 1000 beds hospital, Ayele’s 2017 evaluative study on the use of ceftriaxone in tertiary hospitals showed that around 70% use of antibiotic prophylaxis for a duration exceeding 7 days [26]. Limiting the duration of prophylactic antibiotics is crucial because the use of antibiotics with a longer duration has the potential to change the normal bacterial flora in the patient’s own body and the normal flora in the hospital environment. A disturbed flora can lead to colonization or antimicrobial resistance [25]. The use of prophylactic antibiotics >24 h carries the risk of increasing the incidence of acute kidney injury and infection by *Clostridium difficile* bacteria [27,28].

### 3.3. Profile of Surgical Site Infection (SSI)

Extended preoperative waiting times for hospitalized patients increased the risks of SSI. Preoperative waiting time >7 days could increase the risk of SSI 2.48 times greater than preoperative waiting time <7 days (ARR = 2.48 (95% CI 1.28–4.79); *p* = 0.007) [24]. Both our infected patients underwent surgery after undergoing SSI > 7 days inpatient care. They also had a duration of operation > 1 h. Billoro (2019) found patients who underwent surgery >1 h were at risk of experiencing SSI 2.13 times higher than operations of <1 h (ARR = 2.13 (95% CI 1.18–3.86), *p* = 0.012) [24]. Gachabayov [29] found that the incidence of SSI was significantly higher in diabetic patients with hyperglycemia than nondiabetics ones with hyperglycemia (ARR = 2.10 (1.29–3.42, *p* = 0.002)). Our SSI patients were known to have a record of concomitant diabetes mellitus with preoperative blood sugar levels of 149 mg/dL and 138 mg/dL. Patients with blood sugar level that exceeded 100 mg/dL preincision had shown a 1.7 times greater risk of SSI [30].

The strength of this study is its large number of patients (n > 400) and follow-up. The reported cases of surgical site infection were not when the patient was in the hospital (inpatient) but during 30–60 days after the day of surgery when they returned to the outpatient clinic. The limitation of this study is that severe cases in ICU and those who died were not accounted for, and there were laboratory limitations, and therefore, the bacteria that cause SSI were not identified.

## 4. Materials and Methods

### 4.1. Study Design

A retrospective observational study in both hospitals was conducted by observing patients who underwent surgical procedures during hospitalization at a private hospital in Surabaya (hospital A) and West Nusa Tenggara (hospital B) in the period January–June 2019. Hospital A is a private hospital with 235 beds, whereas hospital B is a government hospital with 411 beds. Both hospitals are referral hospitals. Surgical procedures classifications were (1) clean surgery—surgery performed on an area without preoperative infection and without opening the tract (respiratory, gastrointestinal, urinary, biliary), planned surgery, or primary skin closure with or without the use of closed drains, (2) clean-contaminated operations—operations performed on the ducts (gastrointestinal, biliary, urinary, respiratory, and reproductive except ovaries) or operations without significant contamination, and (3) contaminated surgery—surgery that opened the digestive tract, bile duct, urinary tract, airway to the oropharynx, or reproductive tract except for ovaries with gross spillage [6].

Inclusion criteria included patients aged >18 years old who received prophylactic antibiotics, underwent surgery, and hospitalization in the surgical ward, including surgical wound control within <90 days postoperatively. The exclusion patient criteria were patients in the intensive care unit (ICU), patients who had an infection before surgery, those discharged “against medical advice” (AMA), and those who died.

Slovin’s formula was used to calculate the appropriate sample size (n) from a population using the known population size (N) and the acceptable error value (e). In this study, simple random sampling was used from the population of admitted patients in the study period; N = 267 participants, e = 0.05; therefore, n = 160 participants hospital A; N = 1597 participants, e = 0.05; therefore, n = 320 participants hospital B.

Slovin calculations used Equation (1) as follows:(1)$$n=\frac{N}{\left(1+N\times 0.05^{2}\right)}$$

Antibiotic usage data were obtained from patient medical records and antibiotic use records compiled by pharmaceutical software, which stores all patient data. Data were collected from inpatient medical records, including diagnosis, surgery, length of hospitalization, drug name, dose, and the number of antibiotics received. The dataset was anonymized prior to analysis.

In Indonesia, antibiotics can be prescribed to the patient at admission, during patient care, until the patient’s discharge, and usually two weeks after discharge (next outpatient consultation). In this study, antibiotic use classifications were pre-, during, and postsurgery. Presurgery represents antibiotics given before the day of surgery; antibiotics during surgery are given on the day of surgery (120 min preincision to <24 h after surgery), and antibiotics postsurgery are given >24 h after surgery.

The obtained data were evaluated for antibiotic uses quantitatively and qualitatively. The quantitative evaluation used DDD per 100 bed-days, while the qualitative analysis was the percentage of antibiotic concordance to the American Society of Health-System Pharmacists (ASHP) Therapy Guidelines in (1) route of administration, (2) time, (3) type, (4) dosage, and (5) duration.

Data on bacterial culture test results were obtained from clinical pathology laboratories, and prevention and control committees of infection to follow up regarding the bacteria that cause SSI in postsurgical patients. Patients with SSI were observed in the time they were in the hospital, and when they went to the doctor for review, until the patient was declared cured.

### 4.2. Statistical Analysis

All patient data were anonymized prior to analysis. Data analysis was performed descriptively in the form of an analysis of the quantity of prophylactic antibiotic use with the DDD/100 bed-days, which was calculated using the following formula:(2)$$\frac{\mathrm{DDD}}{100}\mathrm{bed}-\mathrm{days}=\frac{\mathrm{Total}\;\mathrm{Antibiotics}\;\left(g\right)\times 100}{\mathrm{DDD}\;\mathrm{WHO}\;\left(g\right)\times \;\mathrm{LOS}}$$
where DDD WHO = defined daily dose determined by WHO; LOS = total length of stay.

The percentage of compliance with guidelines of antibiotic use with the ASHP therapeutic guidelines was analyzed using the 2014 SIGN criteria as a quality analysis, including the sensitivity profile of the bacteria causing SSI.

A generalized ANOVA model was also carried out on the aggregated DDD dataset to identify the overall difference between the two hospitals in this study. Z-test, in conjunction with Monte Carlo simulation, was also used to investigate which hospital was most adhering to ASHP guidelines. In addition, logit–logistic regression was utilized in this study to further investigate which ASHP adherence components that would be the significant predictor for reported infection cases. Participants’ demographics (e.g., age, gender), comorbidities, and antibiotic usage were included as covariates in the model.

## 5. Conclusions

The usage of prophylactic antibiotics for surgical procedures was high and varied between hospitals. The low concordance to antibiotic therapeutic guidelines suggested injudicious antibiotic use. Further surveillance is needed to identify the incidence of antibiotic-resistant bacteria in the future. It is highly recommended that surgeons abide by guideline recommendations, using only a single antibiotic dose for clean surgery. Further intervention and strategies in implementing antibiotic stewardship are needed to improve antibiotic prescribing and its use.

## Figures and Tables

**Table 1 pharmaceuticals-14-01088-t001:** Demographics information.

Characteristic	Hospital A (N = 164)	Hospital B (N = 323)	Total (N = 487)
Age (years)			
18–30	30 (18.3%)	59 (18.3%)	89 (18.3%)
31–45	56 (34.1%)	95 (29.4%)	151 (31.0%)
46–60	44 (26.8%)	103 (31.9%)	147 (30.2%)
61–75	31 (18.9%)	66 (20.4%)	97 (19.9%)
>75	3 (1.8%)	0	3 (0.6%)
Gender			
Male	61 (37.2%)	158 (48.9%)	219 (45.0%)
Female	103 (62.8%)	165 (51.1%)	268 (55.0%)
Pay providers			
Out of pocket	85 (51.8%)	308 (95.4%)	393 (80.7%)
National Health Insurance	79 (48.2%)	15 (4.6%)	94 (19.3%)
Comorbidities			
Without comorbidities	117 (71.3%)	313 (96.9%)	430 (88.3%)
With comorbidities *	47 (28.7%)	10 (3.1%)	57 (11.7%)
Surgical procedure			
Clean	94 (57.3%)	228 (70.6%)	322 (66.1%)
Clean contaminated	65 (39.6%)	95 (29.4%)	160 (32.9%)
Contaminated **	5 (3.0%)	0	5 (1.0%)

* Comorbidities: diabetes mellitus (DM), hypertension, cardiovascular disease (CVD), gout, cerebrovascular accident (CVA), and others. ** contaminated: an acute incision, nonpurulent drainage.

**Table 2 pharmaceuticals-14-01088-t002:** Antibiotic usage for both hospitals.

Category	Hospital A (N = 164)	Hospital B (N = 323)	Total (N = 487)
The number item of antibiotic			
No antibiotic	15 (9.1%)	0	15 (3.1%)
1 antibiotic	130 (79.3%)	323 (100%)	453 (93.0%)
2 antibiotics	18 (11.0%)	0	18 (3.7%)
>2 antibiotics	1 (0.6%)	0	1 (0.2%)
Injection time given			
0–30 min	35 (21.3%)	57 (17.6%)	92 (18.9%)
31–60 min	32 (19.5%)	83 (25.7%)	115 (23.6%)
61–120 min	55 (33.5%)	46 (14.2%)	101 (20.7%)
>120 min	27 (16.5%)	137 (42.4%)	164 (33.7%)
Record not available	15 (9.1%)	0	15 (3.1%)

**Table 3 pharmaceuticals-14-01088-t003:** The DDD per 100 bed-days antibiotics use for surgical procedures.

ATC * Code	Antibiotic Name	Pre-		During		Post-	
		Hospital A	Hospital B	Hospital A	Hospital B	Hospital A	Hospital B
J01CA04	Amoxicillin	-	7.93	-	-	0.72	1.11
J01CR01	Bactesyn ^a^	-	-	-	5.59	-	-
J01CR02	Clanexi ^b^	-	-	-	-	-	0.54
J01DB04	Cefazolin	0.74	-	2.82	-	1.55	-
J01DB05	Cefadroxil	-	0.94	-	-	-	1.09
J01DD01	Cefotaxime	-	6.07	0.04	8.92	0.04	8.70
J01DD02	Ceftazidime	-	2.87	0.14	3.96	-	3.72
J01DD04	Ceftriaxone	14.71	77.65	22.57	87.31	38.34	93.65
J01DD08	Cefixime	-	18.70	-	-	-	26.79
J01DD62	Cefoperazone Sulbactam	12.26	0.67	3.06	0.62	10.17	0.79
J01DE01	Cefepime	1.47	-	0.14	-	1.45	-
J01DH02	Meropenem	14.34	-	1.59	-	4.94	-
J01FF01	Clindamycin	-	-	-	-	-	1.65
J01GB03	Gentamicin	-	-	-	-	0.17	-
J01MA02	Ciprofloxacin	-	-	-	-	-	1.97
J01MA12	Levofloxacin	-	16.07	0.14	-	0.64	-
J01MA14	Moxifloxacin	-	-	-	23.51	-	3.03
J01XA01	Vancomycin	-	2.68	-	1.60	-	3.41
J01XD01	Metronidazole	0.74	-	2.82	0.89	8.63	2.23
Total		44.26	133.58	33.32	132.40	66.65	148.68

* Anatomical therapeutic chemical; ^a^—ampicillin and beta-lactamase inhibitor; ^b^—amoxicillin and beta-lactamase inhibitor.

**Table 4 pharmaceuticals-14-01088-t004:** The percentage concordance to ASHP Therapeutic Guidelines.

Concordance Type	Hospital A	Hospital B	z-Test (Monte Carlo)
Antibiotic route of administration	96.6	100.0	z = −0.034; *p* < 0.05
Antibiotic injection given time	53.4	42.7	z = 2.211; *p* < 0.05
Antibiotic selection	17.6	1.9	z = 4.966; *p* < 0.001
Antibiotic dosage	16.4	46.7	z = −7.445; *p* < 0.001
Antibiotic dosing interval	19.1	19.2	z = 0.000; *p* = 1.0

**Table 5 pharmaceuticals-14-01088-t005:** The incidence of surgical site infection.

Category	Hospital A	Hospital B
Patients who return for control and culture	0	4
Patients with positive bacteria culture result	0	2
The bacteria culture result	-	Patient 1: *Proteus* sp. dan *S. aureus* Patient 2: *E. coli* dan *S. epidermidis*

## Data Availability

The authors confirm that the data supporting the findings of this study are available on request.

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
