# Peer review of "Discordance to ASHP Therapeutic Guidelines Increases the Risk of Surgical Site Infection"

_pharmaceuticals, 2021, doi:10.3390/ph14111088_

Round 1
Reviewer 1 Report
- In Abstract please state how many hospitals were included in the study.
- In sentence “In the pre-surgery, the timing of antibiotics was given incorrect (45.7%), the dosages were also incorrect” please refrain from the term incorrect, not in accordance with the guidelines would be more appropriate
- Please explain in greater detail the following “Prescribing antibiotics that are not in accordance with the guidelines of therapy can increase infection risk.”
- Prophylactic antibiotics earlier than 120 minutes showed a higher 79 prevalence of SSI than those within 120 minutes (OR 5.26; 95% CI 3.29-8.39) – administered more than 120 minutes prior to the surgical procedure? English language needs thorough editing… please be specific in the sentences you write
- In Methods section please state clearly the number of included hospitals, their characteristics and differences, are both private?
- Also in Methods section is it possible to provide a reference for the classification of surgical procedures?
- Were patients’ data anonymised prior to analysis?
- Please explain your sample size calculation approach, with this approach the results would be applicable only to the two hospitals? what is your population size based on? How was the sampling conducted, was there any clustering to ensure somewhat equal types of surgeries in both hospitals?
- Is it possible to provide a reference for this classification “Antibiotic use classifications were pre-, during, and post-surgery. Pre-surgery represents antibiotics given before the day of surgery; antibiotic use during surgery given on the day of surgery (120 minutes before to…”
- Can patient comorbidities be included in the logistic model? This may also be a cofounding factor.
- Can rationale be provided for such low accordance of selected antibiotic with guidelines in hospital B? maybe of the same manufacturer or something else?
- Timing of the follow up and details about post op infections should be mentioned in the results section also.
- Conclusions should be rewritten to arise from the results of the study and not be an opinion or general standpoint i.e. and will lead to increasing the incidence of bacteria resistant antibiotics – this is not result of the conducted study.
- Consider including some recent research in your discussion and thought the manuscript as the newest references are from 2019…
Author Response
We’d like to thank the Editors and Reviewers for their comments. We have now addressed all reviewers’ comments and made amendments where applicable. Changes can be seen in the manuscript in yellow highlight.
Reviewer 1
In Abstract please state how many hospitals were included in the study.
This information is now added in Abstract; the sentence reads:
“The study subjects were surgical patients from January through June 2019 who met the inclusion criteria as many as 487 participants in two hospitals”
In sentence “In the pre-surgery, the timing of antibiotics was given incorrect (45.7%), the dosages were also incorrect” please refrain from the term incorrect, not in accordance with the guidelines would be more appropriate
We agree with the reviewer, this sentence is now corrected and now reads:
Differences exist in prescribing prophylactic antibiotics pre- and post-surgery. A study showed in the pre-surgery: 46% incorrect timing, 32% incorrect dosages, and 20% too broad antibiotic spectrum, whereas, in post-surgery they found 74% incorrect duration of antibiotic administration, 29% incorrect dosage or frequency, and 6% overly broad spectrum of antibiotics [5].
Please explain in greater detail the following “Prescribing antibiotics that are not in accordance with the guidelines of therapy can increase infection risk.”
This sentence is now corrected and now reads:
Prescribing antibiotics preoperatively can reduce wound infection, whereas not following the guidelines can reduce antibiotics’ effectiveness, for example, different spectrum antibiotics, different tissue antibiotic concentrations at the start and during surgery. Moreover, antibiotic selection pressure leads to antibiotic resistance.
Prophylactic antibiotics earlier than 120 minutes showed a higher 79 prevalence of SSI than those within 120 minutes (OR 5.26; 95% CI 3.29-8.39) – administered more than 120 minutes prior to the surgical procedure? English language needs thorough editing… please be specific in the sentences you write
This sentence is now corrected and now reads:
Prophylactic antibiotics more than 120 minutes pre-incision showed a higher prevalence of SSI than those within 120 minutes (OR 5.26; 95% CI 3.29-8.39) [6-8].
In Methods section please state clearly the number of included hospitals, their characteristics and differences, are both private?
More information has been added and now reads:
Hospital A is a private hospital with 235 beds, whereas Hospital B is a government hospital with 411 beds. Both hospitals are referral hospitals.
Also in Methods section is it possible to provide a reference for the classification of surgical procedures?
Reference 6 added in the end of the sentence which now reads:
…reproductive tract except for ovaries with gross spillage [6].
Were patients’ data anonymised prior to analysis?
Yes, they were; we have now added a sentence in Section 3.2 which reads:
“All patient data were anonymized prior to analysis.”
Please explain your sample size calculation approach, with this approach the results would be applicable only to the two hospitals? what is your population size based on? How was the sampling conducted, was there any clustering to ensure somewhat equal types of surgeries in both hospitals?
We believe that the results from our study is generalised enough, the Slovin formula was used in this study to determine the estimative size that we should sample for. This was done based on the number of participants in each hospital which were 267 and 1597 participants for Hospital A and B respectively. The formula is then applied to ensure that we have a representative of 95% confidence level and that’s how we arrived on 160 and 320 participants for Hospital A and B respectively. The sentence now reads:
In this study, simple random sampling was used from the population of admitted patients in the study period; N = 267 participants, e = 0.05, therefore n = 160 participants Hospital A; N = 1597 participants, e = 0.05, therefore n = 320 participants Hospital B.
Is it possible to provide a reference for this classification “Antibiotic use classifications were pre-, during, and post-surgery. Pre-surgery represents antibiotics given before the day of surgery; antibiotic use during surgery given on the day of surgery (120 minutes before to…”
This has been corrected and now reads:
In Indonesia, antibiotics can be prescribed to the patient at admission, during patient care, until the patient's discharge, and usually two weeks after discharge (next outpatient consultation). In this study..
Can patient comorbidities be included in the logistic model? This may also be a cofounding factor.
We agree with the reviewer – we have forgotten to include this in our original manuscript where comorbidity is included as a covariate, more details has been added which now reads:
“Participants’ demographics (e.g., age, gender), comorbidities, and antibiotic usage were included as covariates in the model.”
Can rationale be provided for such low accordance of selected antibiotic with guidelines in hospital B? maybe of the same manufacturer or something else?
We agree with the reviewer and have added the following sentence:
We postulate that the low accordance in Hospital B may be due to the fact that in Indonesia, antibiotics selection depends on the availability of the drugs rather than guideline recommendation.
Timing of the follow up and details about post op infections should be mentioned in the results section also.
We have added relevant information for this, this reads:
Surgical site infection occurs related to the surgical incision procedure within 90 days follow up
Conclusions should be rewritten to arise from the results of the study and not be an opinion or general standpoint i.e. and will lead to increasing the incidence of bacteria resistant antibiotics – this is not result of the conducted study.
We agree with the reviewer and the Conclusion has been rewritten, this Section now reads
The usage of prophylactic antibiotics for surgical procedures was high and varied between hospitals. The low concordance to antibiotic therapeutic guidelines suggested injudicious antibiotic use. Further surveillance is needed to identify the incidence of anti-biotic-resistant bacteria in the future
Consider including some recent research in your discussion and thought the manuscript as the newest references are from 2019…
We have added relevant references from the latest studies in this manuscript throughout the Discussion section

Reviewer 2 Report
The manuscript covers important subject related to proper use of antibiotics in the patients that undergo surgery. The methods are suitable and clearly described. The results, including the information in the tables are well presented.
The introduction and the discussion contain relevant information. The discussion is well structured and compare the results from the current study with the data from other investigations in proper way. Some sentences should be clarified in the term of English language.
The sentences in the introduction are very long (See lines 62-100 on page 2/13). They should be revised.
The references cited in these lines (62-74) should be cited in different way. Please, remove the titles of the cited studies.
Author Response
We’d like to thank the Editors and Reviewers for their comments. We have now addressed all reviewers’ comments and made amendments where applicable. Changes can be seen in the manuscript in yellow highlight.
Reviewer 2
The manuscript covers important subject related to proper use of antibiotics in the patients that undergo surgery. The methods are suitable and clearly described. The results, including the information in the tables are well presented.
We’d like to thank the Reviewer 2 for the positive comment.
The introduction and the discussion contain relevant information. The discussion is well structured and compare the results from the current study with the data from other investigations in proper way. Some sentences should be clarified in the term of English language.
We agree with the reviewer that some of the sentences did not necessarily flow well together. We have made appropriate amendments throughout this section.
The sentences in the introduction are very long (See lines 62-100 on page 2/13). They should be revised.
A more condensed Introduction is now provided as recommended by the reviewer.
The references cited in these lines (62-74) should be cited in different way. Please, remove the titles of the cited studies.
We agree with the reviewer and appropriate changes have been made.
Some studies reported that the total incompatibility of antibiotic prescribing was high in the surgical ward [3], ICU [3,4], and medical ward [3]; the mismatches to the indication, antibiotic selection [3,4], time/ dose/ route/ duration of antibiotic administration, and different from the Internal Guidelines [3]. There is a discrepancy within the Therapeutic Guidelines in prescribing prophylactic antibiotics pre- and post-surgery. A study showed that in the pre-surgery: 46% incorrect the timing of antibiotics was given, 32% incorrect dosages, and 20% the antibiotic spectrum was too broad, whereas, in post-surgery, 74% incorrect the duration of antibiotic administration, 29% the incorrect dosage or frequency, and there was 6% overly broad spectrum of antibiotics [5].
Added reference: 4. Hussain K, Khan M F, Ambreen G, Raza SS, Irfan S, Habib K, Zafar H. (2020). An antibiotic stewardship program in a surgical ICU of a resource-limited country: financial impact with improved clinical outcomes. Journal of pharmaceutical policy and practice. 2020; 13: 69. https://doi.org/10.1186/s40545-020-00272-w.
This manuscript is a resubmission of an earlier submission. The following is a list of the peer review reports and author responses from that submission.